# DiffusionSeeder: Seeding Motion Optimization with Diffusion for Rapid Motion Planning

**Huang Huang** *
UC Berkeley

**Balakumar Sundaralingam**
Nvidia

**Arsalan Mousavian**
Nvidia

**Adithyavairavan Murali**
Nvidia

**Ken Goldberg**
UC Berkeley

**Dieter Fox**
Nvidia

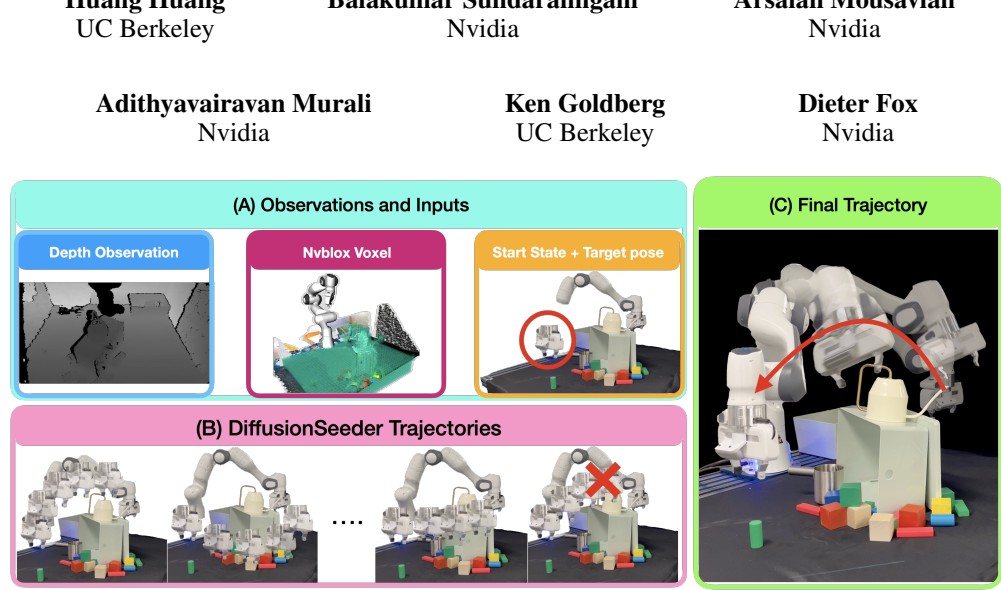

Figure 1: We propose DiffusionSeeder, a diffusion based method that generates diverse, multi-modal trajectories to warm start motion optimization, greatly reducing the planning time on "hard problems".(A) The model takes depth images, the robot start configuration, and target pose. (B) DiffusionSeeder generates diverse multi-modal trajectories, where some trajectories could be in collision.A dynamic Euclidean Signed Distance Field (ESDF) using nvblox [1] from the depth images is used by the motion optimizer cuRobo to move trajectories out of collision. (C) The final output from the motion optimizer is smooth, fast, and collision-free.

**Abstract:** Running optimization across many parallel seeds leveraging GPU compute [2] have relaxed the need for a good initialization, but this can fail if the problem is highly non-convex as all seeds could get stuck in local minima. One such setting is collision-free motion optimization for robot manipulation, where optimization converges quickly on easy problems but struggle in obstacle dense environments (e.g., a cluttered cabinet or table). In these situations, graph-based planning algorithms are used to obtain seeds, resulting in significant slowdowns. We propose DiffusionSeeder, a diffusion based approach that generates trajectories to seed motion optimization for rapid robot motion planning. DiffusionSeeder takes the initial depth image observation of the scene and generates high quality, multi-modal trajectories that are then fine-tuned with a few iterations of motion optimization. We integrate DiffusionSeeder to generate the seed trajectories for cuRobo, a GPU-accelerated motion optimization method, which results in 12x speed up on average, and 36x speed up for more complicated problems, while achieving 10% higher success rate in partially observed simulation environments. Our results show the effectiveness of using diverse solutions from a learned diffusion model. Physical experiments on a Franka robot demonstrate the sim2real transfer of DiffusionSeeder to the real robot, with an average success rate of 86% and planning time of 26ms, improving on cuRobo by 51% higher success rate while also being 2.5x faster. More information are in the website.

**Keywords:** Robot Motion Planning, Diffusion Model

---

*Work done during internship at NVIDIA

8th Conference on Robot Learning (CoRL 2024), Munich, Germany.

# 1   Introduction

Robot motion generation aims to find a collision-free path in robot configuration space to reach a desired goal pose from a starting configuration. It is challenging due to the high dimensional robot configuration space, where each configuration corresponds to a combination of robot joint space and an environment representation. This problem has been studied using sampling-based methods [3, 4], which sample configurations in the configuration space and checks for collisions in-between until a complete path is found. Others have leveraged optimization-based methods [5, 6, 7], which optimize an initial path under specific objectives and constraints, to solve this problem. For both approaches, planning time increases significantly with the complexity of the configuration space [8], resulting in a large variance in planning time for different problems. Furthermore, sampling-based methods often require post-processing to smooth the generated paths, while optimization-based methods, which can generate smooth paths directly, are highly sensitive to the initial seed. Several optimization-based algorithms [2, 9, 5] address this problem by starting from a set of seed trajectories for optimization. The quality of seed trajectories are crucial, as an absence of good seeds can lead to failure in densely populated environments. Therefore, modern implementations of planners are forced to resort to graph-based planning algorithms to guarantee completeness at the cost of speed.

Recently, learning-based methods have demonstrated great promise at motion generation without access to ground-truth state, achieving faster planning than sampling-based methods [9, 10, 11]. Some work [8, 12, 11] predict the next joint state iteratively instead of the entire trajectory, increasing the planning time. Learning-based methods often lack theoretical guarantees, fail to generalize to out of distribution scenes, and tend to generate sub-optimal plans as we later show in our results.

We aim to develop a rapid and practical off-the-shelf motion planning pipeline that works with partial observations. We propose DiffusionSeeder, which generates diverse, high-quality paths to seed the optimization based motion planner, as shown in Figure 1, significantly reducing both the seed trajectory generation time and the number of optimization steps required. Denoising Diffusion Probabilistic Models (DDPM) [13], a class of generative networks that model the output generation as a denoising process, have shown success in capturing complex multi-modal distributions in image generation and, recently, in robot manipulation [14]. Robot motion planning is well known to be a multi-modal problem, making DDPM a great fit for seed trajectories generation. DiffusionSeeder uses a conditional DDPM, consisting of an observation encoder and a noise prediction network for seed trajectory generation. We represent the partially observed environment with depth observations, which are faster to process than point clouds [11, 8, 12]. We train DiffusionSeeder on 3M simulated scenes from M$\pi$Net [11]. We use cuRobo, a fast motion optimizer utilizing GPU hyper-parallelization, to optimize DiffusionSeeder generated seeds for final trajectories.

This paper makes the following contributions:

- DiffusionSeeder, a conditional DDPM that generates high-quality seed trajectories within 6 milliseconds from a raw depth image, camera pose, start joint configuration and end pose.
- A robot motion planning dataset consisting of 15M smooth and fast trajectories generated by cuRobo on 3M M$\pi$Net [11] scenes in simulation.
- Integration of DiffusionSeeder with cuRobo, achieving $12\times$ average speed up over standard cuRobo (and 36x speed up at 98th quantile), while achieving a 10% higher success rate on partially occluded views of a scene.

# 2   Related Work

Motion planning for robotics has been broadly explored. Search-based planning methods such as A* [15, 16] are complete, fast, and optimal for problems with discrete state spaces. Sampling-based methods such as RRT [3], and PRM [4] are asymptotically optimal but can generate sub-optimal and erratic trajectories in practice with limited compute budget. Optimization-based methods [5, 6, 7] are susceptible to local minima in complex problem spaces. GPU-parallelized computations have been leveraged to address this local minima issue [2]. cuRobo [2] is a GPU-accelerated motion

optimizer that solves on many seeds with linear interpolations from start to goal configurations in parallel to avoid local minima. In the case of failure, it falls back to a graph based motion planner to generate new seeds which significantly slows down the planning time, taking up to 0.5 seconds. Our work aims to mitigate this issue by using a diffusion model to generate high quality multi-modal seed trajectories, resulting in fast convergence and removing the need for graph based planner.

Many works have utilized a learned neural network for generating environment-dependent sampling distributions [17, 9, 10, 18, 19], accelerating sampling-based planning by biasing samples toward collision-free joint configurations. But the planning time still scales with the environment complexity. Other approaches learn a collision model [20, 21] for fast collision checks, but still rely on exhaustive sampling with Model Predictive Path Integral (MPPI) [22, 23] and a local controller. Other works [11, 8, 12] have formulated the motion planner as a policy which is queried sequentially to generate a trajectory. Motion Planning Networks (MPNet) [8, 12] and M$\pi$Net [11] learn policies to directly predict the next joint configuration from a point cloud. However, motion planning policies require multiple rollout steps to generating one trajectory, resulting in longer planning time. There is no further optimization on the generated trajectory, which can result in suboptimal trajectories. Some approaches, such as Ichnowski et al. [24], warm-start optimizers with network-generated seeds, but lack generalization to new scenes and output only one trajectory. Yoon et al. [25] learn a reinforcement learning policy to generate an initial trajectory, but multi-step rollouts are slow and limited to one result, while DiffusionSeeder can generate multiple diverse trajectories quickly.

Diffusion models have been leveraged for trajectory optimization. Janner et al. [26] trains a diffusion model per environment, making it impractical for diverse scenes. Carvalho et al. [27] and Saha et al. [28] train an observation/environment agnostic diffusion model and leverage the problem costs to guide the diffusion sampling to generate trajectories priors satisfying the scene-specific constraints, which may struggle to generate collision-free trajectories in partially observed cluttered environments. DiffusionSeeder is conditioned on an environment encoding and can generate environment-specific trajectories for various environments, which are more efficient for motion planning in complicated environments. Huang et al. [29] represents the environment as point clouds and utilizes goal-oriented planning for robot motion planning, formulated as a motion in-painting problem, which requires multiple policy rollout steps for generating one trajectory. DiffusionSeeder takes in depth observations as the input to a Vision Transformer (ViT), in contrast to slower-to-process point clouds. Further, DiffusionSeeder generates multiple full trajectories for each inference pass of the diffusion model, avoiding multi-step policy rollouts.

## 3   Problem Statement

We study the robotics motion planning problem. The goal is to generate a fast, smooth and collision-free trajectory $\tau$ to bring a robot from a starting configuration to a goal pose, given a depth image of dimension $H \times W$, the calibrated intrinsic and extrinsic matrices of a third-person view camera, the starting robot joint state, $q_0$, and the goal robot end effector pose (a 7d vector of a translation and quaternion in Cartesian space). We aim to develop a seeder that can generate diverse and high quality seed trajectories to accelerate a motion optimizer to rapidly generate a collision-free trajectory $\tau$ with position and rotation errors below given thresholds ($\delta_t$ and $\delta_r$, respectively), while minimizing the jerk and motion time of the generated trajectory. We define $\tau$ to be $\{(t_i, q_i)|i \in [0, T]\}$, where $t_i$ is the time step, $q_i$ is the robot joint state at time step $t_i$, and $T$ is the length of the trajectory.

## 4   Method

### 4.1   Dataset Generation

We generate our dataset using cuRobo for a 7-DoF Franka robot on 3M M$\pi$Net training scenes [11] in three categories: cubby, tabletop and dresser (Appendix Fig 5). A motion planning problem in the M$\pi$Net dataset consists of a scene mesh geometry, a start and end pose. To increase data diversity, we sample additional pairs of start and end poses based on the original start and end poses. Problems for which cuRobo is unable to find a feasible solution are discarded. We generated 15M problems

with feasible and smooth solutions of size 32×7 on 3M scenes, where 32 is the trajectory length $T$ and 7 is the robot DoF. More details are in Appendix 7.1.

To provide partial observations to the planner, we render depth images for each scene using a fixed virtual camera located within the scene. To generalize to different view points, we sample 12 different camera poses for each scene. We render depth images from each camera pose and filter out camera poses where the problem end pose is not visible in the rendered depth images, resulting 12M rendered depth images of size $256 \times 256$ on 3M scenes. We save the corresponding camera intrinsic and extrinsic in robot base frame for each depth image. More details are in Appendix 7.2.

## 4.2  Model Architecture

DiffusionSeeder uses a conditional DDPM, consisting of an environment observation encoder and a conditional noise prediction network, to generate diverse seed trajectories, which can be optimized by an optimizer cuRobo for the final trajectory, as shown in Figure 2.

The observation encoder processes depth images, camera poses, the start joint configuration, and the goal pose to create a single embedding vector as the condition of the denoising diffusion process. Specifically, we use a ViT with 12 layers and 12 heads as the network backbone. Depth images, projected to 3D using camera intrinsics, are divided into 64×64 patches where each patch is encoded first and concatenated with the linear projection of the $SE(3)$ homogeneous transformation of each camera pose. We add a positional embedding to each patch embedding, resulting a single 512-dimensional visual embedding using the class token of ViT. The visual embedding is concatenated with the 64-dimensional encoding vector of the start configuration and the goal pose, resulting in a 576-dimensional environment embedding vector for the conditional noise prediction model.

We adopt the CNN based architecture from [14] as the conditional noise prediction model, which is a 3-level UNet [30] architecture consisting of conditional residual blocks with channel dimensions $[256, 512, 1024]$. The model encodes the time step into a latent vector of 256 dimensions through a multi-layer perceptron (MLP), which is concatenated with the environment embedding from the observation encoder, resulting an 831-dimensional conditional vector.

## 4.3  Model Training and Inference

We jointly train the observation encoder and noise prediction model of the DDPM. During training, for each problem, we sample one depth image from the pre-rendered depth images. At each training iteration, we sample a ground truth trajectory $\tau$ from the dataset. We randomly select a denoising step $k \in [0, K]$ and a random noise $\epsilon$, which is added to the ground truth trajectories, resulting a noisy trajectory $\tilde{\tau} = \tau + \epsilon$. The objective of the noise prediction network $\Theta$ is to predict the noise added to the original trajectory, with the training loss defined as

$$L = MSE(\epsilon, \Theta(\tilde{\tau}, k, \Phi(O))), \tag{1}$$

where $\Phi$ is the observation encoder and $O$ is the observation.

In robot motion planning problems, the error in each joint can have different impact on the end effector error, due to the non-linearity of the robot forward kinematics (FK) model. To mitigate this issue, we pass both $\epsilon$ and the predicted noise $\hat{\epsilon}$ to the robot FK model to reflect this non-linearity. We use FK to obtain the position of many points sampled on the robot's geometry across all links. We calculate the distance between the predicted positions of these points and the labels (using FK on joint state) as the loss. This loss gives a more direct representation of the error in the Cartesian Space for the whole robot.   In addition, as many of the motion planning problems can be solved through a linear interpolation solution, we upweight the loss for non-linear solution to encourage the model to pay attention to the non-linear solutions, which are often harder to solve. As cuRobo will only call a graph based planner when the linear interpolation solution has failed, we use this to approximate the non-linearity of the trajectories. Together, we define our training loss as

$$\mathcal{L} = (1 + \alpha \mathbb{1}(\tau)) MSE(FK(\epsilon), FK(\Theta(\tilde{\tau}, k, \Phi(O)))), \tag{2}$$

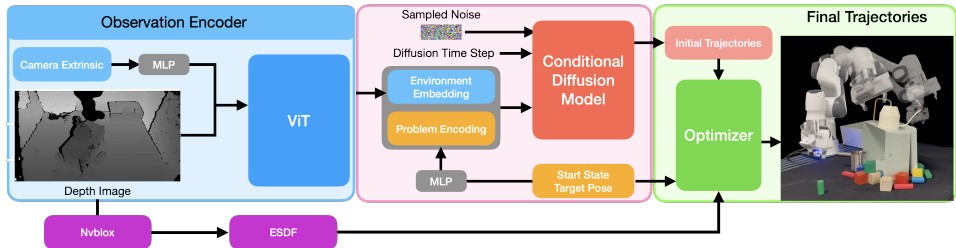

Figure 2: **DiffusionSeeder trajectory generation pipeline**. DiffusionSeeder consists of a ViT observation encoder (blue) and a conditional diffusion model (coral). The robot start configuration and target pose are embedded through an MLP into a "problem encoding", which is concatenated with the environment embedding. The diffusion model conditions on the concatenated vector and a randomly-sampled noise tensor of size $32 \times 7$ to generate initial trajectories for the optimizer (green). The optimizer takes in the ESDF for collision checking. The final trajectory after optimization is shown on the right.

where $\mathbb{1}(\tau)$ is an indicator of whether the ground truth trajectory $\tau$ is generated from graph based planner and $\alpha$ is a scale factor. We train our model with depth images of size $256 \times 256$, $K = 100$, $\alpha = 4$ and a batch size of 256 for 72 epochs, which takes 2 days on 8 NVIDIA A100 GPUs.

We use Denoising Diffusion Implicit Models (DDIM) [31] for faster inference. Combined with CUDA optimization, we achieve a 6ms inference time on an Nvidia GeForce RTX 4090, which includes the inference of the vision encoder and 5 steps of denoising.

## 4.4 Optimization

We incorporate the trained DiffusionSeeder with an existing trajectory optimizer available in cuRobo. Joint space trajectories $\tau$ from DiffusionSeeder, the robot start configuration $q_0$, target end pose $X_d$ are passed to the optimizer to optimize for $N_{\text{iters}}$ iterations. Details are in Appendix 7.6.

In partially observed scenes, for optimizer collision checking, we construct a Euclidean Signed Distance Field (ESDF) using nvblox [1] from input depth images. Nvblox is a GPU-accelerated signed distance field construction library designed for robotic path planning, which uses GPU parallel computation to efficiently perform volumetric mapping of the world, while also enabling sharing of the generated map in a zero-copy mode with cuRobo for trajectory optimization on the GPU.

| Metric | BiTStar | | M$\pi$Net | | cuRobo v0.6.2 | | | DiffusionSeeder | | | | |
|---|---|---|---|---|---|---|---|---|---|---|---|---|
| Condition | $\tilde{\delta}$ | $\delta$ | $\delta'$ | $\delta$ | $N_{atp} = 1$ | 10 | 100 | $N_{iters} = 25$ | 50 | 100 | 200 | 475 |
| Plan Time (s) | 0.52 | 0.48 | 0.50 | 1.95 | 0.049 | 0.082 | 0.207 | **0.015** | 0.017 | 0.020 | 0.027 | 0.045 |
| Total Time (s) | 0.69 | 0.65 | 0.50 | 1.95 | 0.079 | 0.112 | 0.237 | **0.045** | 0.047 | 0.050 | 0.057 | 0.075 |
| Success Rate | 26.6% | 6.0% | 27.4% | 8.3% | 66.2% | 77.7% | 77.9% | 85.1% | 85.8% | 84.9% | 85.1% | **86.2%** |
| Jerk ($rad/s^3$) | **47.2** (81.4) | 49.9 | 56.8 | 60.6 | 98.5 | 96.7 | 97.8 | 108.9 | 103.6 | 99.3 | 93.5 | 89.6 |
| Motion Time (s) | 1.84 (1.72) | 1.98 | 5.35 | 7.71 | **1.14** | 1.17 | 1.18 | 1.26 | 1.26 | 1.27 | 1.30 | 1.26 |
| Translation Err (mm) | 3.89 | 4.05 | 8.66 | 3.92 | **0.05** | 0.06 | 0.06 | 0.98 | 0.95 | 0.91 | 0.50 | 0.78 |
| Quaternion Err (°) | 13.3 | 1.10 | 7.27 | 2.68 | **0.63** | 0.90 | 0.93 | 1.78 | 1.44 | 1.20 | 1.03 | 0.92 |

Table 1: Evaluation with a partial observation of a depth image. Mean values of each metric on successfully solved problems over the 1791 test problems are reported. As DiffusionSeeder-50 has a similar success to DiffusionSeeder-475 but is 60% faster at planning, we use DiffusionSeeder-50 when reporting primary results. The jerk and motion time of re-timed BiTStar trajectories are shown in the parenthesis.

## 5 Experimental Evaluation

We conduct experiments both in simulation and on a physical Franka Panda robot to evaluate DiffusionSeeder. In the following sections, we denote DiffusionSeeder as DiffusionSeeder combined with cuRobo, unless otherwise specified. We use the same set of parameters for both simulation and real-world experiments. We run DiffusionSeeder with $N_{\text{denoising}}$ denoising steps to generate trajectories of length $T = 32$ in the joint space. For each problem, we pass $N_{\text{trajs}}$ randomly sampled noise with the same condition from the observation encoder to the diffusion model to generate $N_{\text{trajs}}$ initial trajectories for optimization. $N_{\text{denoising}} = 5, N_{\text{trajs}} = 12$ achieves good trade off between generation quality, diversity, and inference time. We consider the following quantitative metrics: success rate, plan time, jerk, translation error $\delta_t$, quaternion error $\delta_r$, and motion time (See Appendix 7.7). All metrics are computed over successful trajectories.

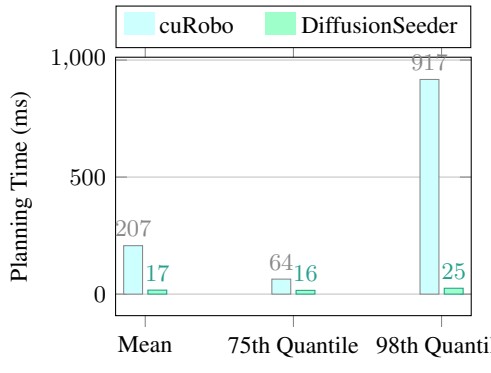
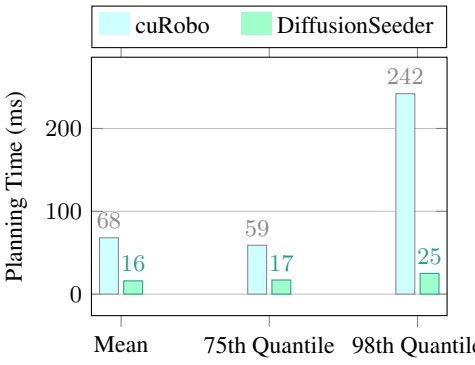

(a) Planning in partial observed environments.  (b) Planning in fully observed environments.

Figure 3: Planning time for cuRobo-100 and DiffusionSeeder-50 in partially and fully observed environment on the MπNet test set [11]

## 5.1 Simulation Experiments

We evaluate DiffusionSeeder on the MπNet simulation test set of 1791 problems. Each problem has a scene from the same scene types as the training data but with different configurations (Figure 5). We compare DiffusionSeeder to BiTStar, MπNet and cuRobo(v0.6.2). We use $\delta_t = 0.005m$ and $\delta_r = 2.86°$ as the success threshold for both cuRobo and DiffusionSeeder as in [2].

We represent scenes with a single depth image. Both cuRobo and DiffusionSeeder use nvblox with the given depth image for collision checking, which takes 30ms to reconstruct the ESDF. The total time of DiffusionSeeder and cuRobo is the sum of the planning and nvblox reconstruction time. In practice, this can be expedited by running nvblox construction in parallel with the diffusion model inference. For BiTStar, mesh generation from the depth image takes 0.17s. For MπNet, we project depth images to point clouds using the camera intrinsics. We evaluate the generated trajectories from all method using the ground truth mesh for collision checking (see Appendix 7.8).

We run cuRobo for a maximum of $N_{atp}$ attempts, denoted as cuRobo-$N_{atp}$. For each attempt, we check if the trajectory meets the success criterion, repeating until a successful trajectory is generated or the attempt limit is reached. We use $N_{atp} = 1, 10, 100$ as more attempts does not further increase the performance. For DiffusionSeeder, we run $N_{iters}$ optimization iterations for the diffusion generated trajectories, denoted as DiffusionSeeder-$N_{iters}$. We use $N_{iters} = 25, 50, 100, 200, 475$, as more optimization iterations doesn't increase the trajectory quality much but increases the planning time. We evaluate all methods on a Nvidia GeForce RTX 4090.

The average values of each metric are summarized in Table 1. Experiments in fully observed environments using ground-truth meshes, ablations on $N_{trajs}$ and number of depth images, DiffusionSeeder only without cuRobo and more discussions are included in Appendix 7.8.

### 5.1.1 DiffusionSeeder vs Sampling Based Planner

BiTStar [32] is a sampling-based planner that utilizes the benefits of the random geometric graph. It can generate high quality and optimal solutions quickly compared to other sampling-based methods, especially in high dimensional problems, and has the properties of probabilistic completeness and asymptotic optimality. We incorporate BiTStar from OMPL [33] into MoveIt2 [34] with a timeout of 5s and a maximum of 10 attempts per timeout. Since MoveIt2 constructs goal constraints using the Euler angle error on each axis, we set the angle error tolerance on goal pose to be 0.01 rad for each axis and the translation error tolerance to be 0.005 m. We denote this success threshold as $\tilde{\delta}$. We report the performance of BiTStar on both $\tilde{\delta}$ and $\delta$. Table 1 shows that DiffusionSeeder has a success rate 3x higher and a planning time 3% of that of BiTStar under the less strict success threshold $\tilde{\delta}$. The low success rate of BiTStar provides a benchmark for the difficulty of the motion planning problems under partial observations. BiTStar has a lower jerk compared to DiffusionSeeder

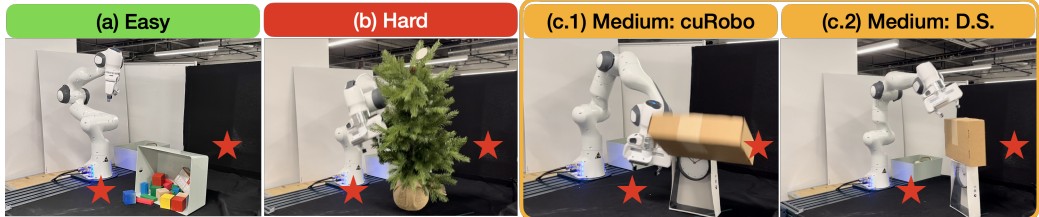

Figure 4: Physical experiments scenes: (a) an easy scene with a box, (b) a hard scene where both cuRobo and DiffusionSeeder failed, (c) a medium scene. (c.1) shows cuRobo colliding while (c.2) shows DiffusionSeeder avoiding collision in the same scene. The start and end position are marked by the red stars. The two positions are to the left and right of the obstacles, requiring the robot to navigate around the obstacle for a safe execution.

as DiffusionSeeder generates faster trajectories, indicated by the lower motion time. When we re-timed the trajectories from BiTStar so that trajectories reach the robot's velocity or acceleration limits, the maximum jerk of BiTStar generated trajectories became 81.4 $rad/s^3$ with a motion time of 1.72s. BiTStar generated trajectories have 21% lower jerk while also being 27% slower than DiffusionSeeder generated trajectories.

### 5.1.2 DiffusionSeeder vs Prior End-to-End Approach

M$\pi$Net [11] is an end-to-end method that takes point clouds and directly predicts waypoints in the configuration space. We use the best pre-trained model (M$\pi$Net trained on the Hybrid Expert) as our baseline to compare with. We use a maximum rollout number of 150 for M$\pi$Net as in [11]. Additionally, we report the success rate of M$\pi$Net under the original success threshold of [11]: $\delta' = (\delta_t = 0.01m, \delta_r = 15°)$. Table 1 shows that DiffusionSeeder outperforms M$\pi$Net by an order of magnitude in terms of success rate (8.3% vs 85.8%). We hypothesize that this substantial improvement comes from combining the power of a learning approach with a classical motion planner. Since M$\pi$Net is a policy that predicts one-step delta joint configuration, it would be significantly slower to combine a multi-step M$\pi$Net trajectory (unrolled in a auto-regressive way) with classical methods for reactive control. However, M$\pi$Net allows the robot to start moving after the first inference is completed ($\sim$7 ms), while DiffusionSeeder always gives a complete trajectory in 17 ms.

### 5.1.3 DiffusionSeeder vs Heuristic Seed Generation

cuRobo uses heuristics to generate seed trajectories and its success rate depends heavily on the quality of seed trajectories it uses to optimize. At each attempt, cuRobo samples a new batch of seeds. From Table 1, the success rate of cuRobo increases monotonically with $N_{atp}$, indicating the inefficiency of its heuristic sampling as it needs to sample $N_{atp}$ times to achieve a high success rate, while DiffusionSeeder combined with cuRobo achieves higher performance in one attempt. Another effect of seed trajectories is on the number of optimization iterations. We set $N_{iters} = 475$ for cuRobo to optimize the seed trajectories, which was shown to achieve the highest success rate for cuRobo. DiffusionSeeder can achieve higher performance with significantly fewer iterations ($N_{iters} = 25$), suggesting the generated seed trajectories are closer to the optimal collision-free trajectories.

### 5.1.4 Planning Time Comparisons

Both cuRobo and DiffusionSeeder outperform M$\pi$Net and BiTStar by a large margin in planning speed. DiffusionSeeder-50 and cuRobo-1 are 11x and 6x faster than M$\pi$Net with success threshold $\delta'$, respectively, in terms of total time. Since both cuRobo and DiffusionSeeder use nvblox for ESDF reconstruction, we compare the planning time between cuRobo and DiffusionSeeder instead of the total time. On average, DiffusionSeeder-50 is 3x faster than cuRobo-1 with a 30% higher success rate. Compared to best-performing cuRobo-100, DiffusionSeeder-50 is 12x faster with a 10% higher success rate. We show the 75th and 98th quantile of the planning time of cuRobo-100 and DiffusionSeeder-50 in both partially and fully observed environments (Appendix 7.8.6) in Figure 3, where cuRobo planning time has a great discrepancy across the mean, 75th quantile, and 98th quantile. DiffusionSeeder-50 is 4x faster than cuRobo-100 on the 75th quantile and 36x faster

| Scenes | Success | | Motion Time (s) | |
|---|---|---|---|---|
| | cuRobo | DiffusionSeeder | cuRobo | DiffusionSeeder |
| Empty | 5/5 | 5/5 | 34.6 | 40.4 |
| Easy | (5/5,5/5) | (5/5,5/5) | 39.5 | 43.3 |
| Medium | (0/5,0/5) | (5/5,5/5) | - | 41.5 |
| Hard | (5/5,0/5) | (5/5,0/5) | 40.8 | 44.1 |
| Mean | 57% | 86% | 38.0 | 42.3 |

Table 2: Real experiment results for cuRobo and DiffusionSeeder. Each difficulty tier contains two scenes with 5 trials repeated for each scene. We report number of successful trials for each scene in each tier. The method is successful if it doesn't collide with the environment across all trials in a scene.

on 98th quantile in partially observed environments. While cuRobo shows high variance in planning time, DiffusionSeeder achieves relatively consistent performance among all scenes, indicating its advantage in generating scene-specific seeds.

## 5.2 Real Robot Evaluation

We conduct experiments on a Franka Panda robot across 6 scenes, categorized into 3 difficulty tiers with 2 scenes from each tier and also an empty scene. Some of these scenes are shown in Figure 4. None of the environment setups are part of our training dataset. In each scene, we select three poses that the robot needs to reach sequentially, repeating 5 times. A method is considered successful if it avoids collisions across all 5 trials in a scene. In addition to success, we also report the time the robot was executing trajectories as Motion Time. More details are in Appendix 7.9.

From Table 2, DiffusionSeeder failed once among the 7 scenes with an average success rate of 86% while cuRobo failed 3 times with an average success rate of 57%, demonstrating the sim2real transfer of DiffusionSeeder. DiffusionSeeder outperforms cuRobo on planning time with an average planning time of 26ms while cuRobo has a planning time of 65ms. The motion time of the planned trajectories of DiffusionSeeder is higher than that of cuRobo, similar to that in simulation experiments. As discussed in Appendix 7.8.5, we hypothesize this may be attributed to the additional weight $\alpha$ on loss for non-linear trajectories, resulting DiffusionSeeder to generate more non-linear trajectories, which are less likely to collide but also have higher motion time. All failures for cuRobo was due to limited view of the obstacles from a single camera. The one environment where DiffusionSeeder failed was because the tree obstacle did not fully fit in the view of the camera.

## 6 Conclusion and Limitation

We propose DiffusionSeeder, a diffusion-based model for generating initial seed trajectories for motion planning in novel scenes from just depth observations. Integrated with cuRobo, it achieves up to 36× speedup on complex motion-planning tasks. DiffusionSeeder is trained on a large-scale synthetic dataset but shows generalization to unknown real world scenes and observations. By utilizing an environment encoder and a broad set of views of each scene during training, DiffusionSeeder generates collision-free seed trajectories under partial observation. In future, we hope to explore the capabilities of the environment encoder for single-view scene geometry understanding and investigate its applicability in other robotic domains, such as grasping and manipulation.

DiffusionSeeder has a few limitations. It generates trajectories based on a fixed external camera view, making it less effective when the goal pose is occluded. We hope to extend DiffusionSeeder to incorporate multiple input views and more diverse camera poses in the future, to benefit from multi-camera or wrist-mounted camera setups. DiffusionSeeder is only trained on the Franka robot. We hope to extend DiffusionSeeder to different robots in the future. While we empirically observed that the nonlinearity introduced in Eq. 2 improves the performance, there is a lack of thorough analysis on the effectiveness of Eq. 2 compared to a regular MSE loss. A comprehensive study on the loss function design for robotic motion planning can be an interesting future direction.

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

# 7 Appendix

## 7.1 Problem Generation

To increase data diversity, we sample additional pairs of start and end state poses near the original start and end poses. Since most scenes have obstacles near the end pose, we also sample problems near the end pose to encourage difficult trajectory generation. Specifically, for each scene, we obtain the start pose in Cartesian space from the M$\pi$Net start state and sample N-1 new motion planning problems (start and end pose pairs) within a bounding box of 0.3m along the X- Y- and Z-axes around the original pose. We sample N new problems within a bounding box of 0.3m near the end pose. We also sample N pairs of poses in free space, within a cuboid of length 0.6m, to increase our coverage over the joint configuration space of the robot. This results in 3N start and end pose pairs for each scene. We run inverse kinematics for each pose in the problem to filter out ones where either pose is in collision with the scene.

We pass each motion planning problem to cuRobo to generate smooth and collision-free trajectories with a fixed trajectory length $T = 32$. We define the trajectory length as the number of waypoints and perform linear interpolation between the points. Problems for which cuRobo is unable to find a feasible solution are filtered out and discarded. With $N = 5$, we generated 15M problems with feasible and smooth solutions of size 32×7 on 3M unique scenes.

## 7.2 Environment Rendering

Each camera pose is represented by an azimuth $a$, an elevation $e$, and a radius $r$. For each scene, we discretize orientations to a fixed grid of $M = N_a \cdot N_e$ (where $N_a$ are discretization resolution on azimuth and $N_e$ is discretization resolution on elevation) camera poses on a sphere centered at a scene mesh and pointing inward toward the center of the scene. Azimuths range is $[-\pi/4, \pi/4]$, elevations range is $[\pi/5, 2\pi/5]$, and radius $r$ range is $[0.7 \times R, 1.1 \times R]$ where $R$ is the radius of scene mesh. We apply uniform random noise $\epsilon_a \in [-\pi/8, \pi/8]$ and $\epsilon_e \in [-\pi/15, \pi/15]$ to each discretized $a, e$ respectively to obtain diverse camera poses across different scenes.

We render depth images from each camera pose and filter out camera poses where the problem end pose is not visible in the rendered depth images. The goal is visible if the projection of the goal in the camera view has larger depth than the goal itself—in other words, none of the observed pixels can occlude the goal in the camera frame. Using $N_a = 4, N_e = 3$ and saving up to 4 images per scene (randomly sampled from the valid ones), we rendered 12M depth images of size $256 \times 256$ on 3M scenes.

## 7.3 Denoising Diffusion Probabilistic Models

A DDPM parameterized by $\theta$ denoises a given $x_K$ for K iterations to generate $x_0$, where $x_0 \sim q(x_0)$. $x_{1:K}$ are generated through a diffusion process $q(x_k|x_{k-1})$, modeled as a Markov chain, which gradually adds Gaussian noise to $x_0$. The training objective of a DDPM is therefore to minimize the negative log likelihood $E[-\log p_\theta(x_0)]$, which is upper bounded by [13]

$$E_q[-\log p_\theta(x_K) - \sum_{t \geq 1} \log \frac{p_\theta(x_{k-1}|x_k)}{q(x_k|x_{k-1})}], \tag{3}$$

As the upper bound is calculated in expectation of the diffusion process $q(x_k|x_{k-1})$, we sample different time steps $k \in [0, K]$ during the training for estimating the expectation. The denoising process of DDPM is defined as a Markov chain with a learned Gaussian transitions such that

$$p_\theta(x_{k-1}|x_k) := \mathcal{N}(x_{t-1}; \mu_\theta(x_k, k), \Sigma_\theta(x_k, k)) \tag{4}$$

As shown in [13], the parameterization of the denoising process can be modified to train a noise prediction model to predict the noise from $x_k$.

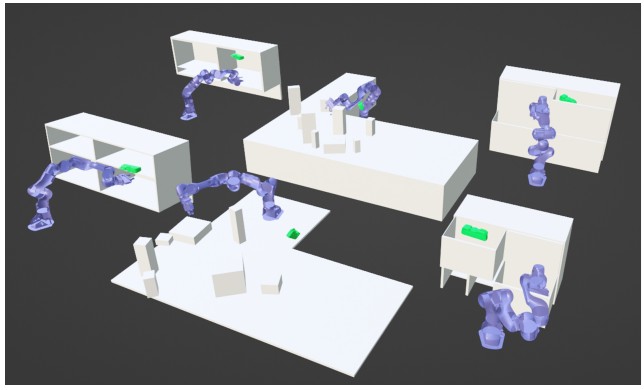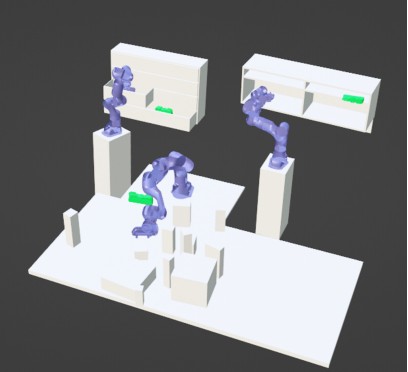

Figure 5: Example train (left) and test (right) problems on three scene types: cubby, tabletop, and dresser (from left to right in the left image). The robot is in the start configuration and the target end pose is marked in green.

## 7.4 Model Training

During training, for each problem, we sample one depth image from the pre-rendered depth images. We also reverse the start and end pose, and the order of the ground truth trajectories half of the time to generate problems going from the end to the start, for further data augmentation. The ground truth trajectories and the input robot start configuration are normalized to be from 0 to 1 by dividing the joint position by the corresponding joint mechanical limit, to improve the training stability. We then denormalize the output of the diffusion model by multiplying with the joint mechanical limit.

## 7.5 Model Inference

As the diffusion and denoising process of a DDPM are both Markovian, DDPMs generate $x_0$ by denoising $K$ steps from $x_K$, resulting in longer inference time. Denoising Diffusion Implicit Models (DDIM) [31] are an efficient class of iterative implicit probabilistic models. They have the same training objective as DDPMs but model non-Markovian diffusion processes. DDIMs thus allow the diffusion model trained with DDPM to generate $x_k$ directly from $x_{k+m}$ without the generation of the intermediate steps, speeding up the inference time by greatly reducing the number of denoising iterations. During inference time, we randomly sample a noise vector of the same shape as the trajectory, and use a DDIM to generate a denoised trajectory of shape 32×7. We convert the trained model to BFLOAT16 for accelerated inference and implement the DDIM step as a fused CUDA kernel leveraging JIT scripting in Pytorch 2.0. We encode the full inference step in a single CUDA Graph to further reduce python overheads. These techniques give us a 6ms inference time on an Nvidia GeForce RTX 4090, which includes the inference of the vision encoder, followed by 5 steps of denoising.

## 7.6 Optimization Problem

We use the trajectory optimization problem formulated in cuRobo [2], which we write briefly below,

$$\min_{\tau} \quad C_{\text{goal}}(X_d, FK(q_T)) + C_{\text{smooth}}(\tau)$$
$$+ C_{\text{self}}(\tau) + C_{\text{world}}(\tau)$$
$$\text{s.t.} \quad \text{joint limits}$$

where $C_{\text{goal}}$ is the cost for reaching the goal pose at the final timestep $T$, $C_{\text{smooth}}$ is a cost term to encourage smooth minimum-jerk trajectories, $C_{\text{self}}$ and $C_{\text{world}}$ are self and world collision avoidance cost terms.

## 7.7 Metrics

We consider the following quantitative metrics:

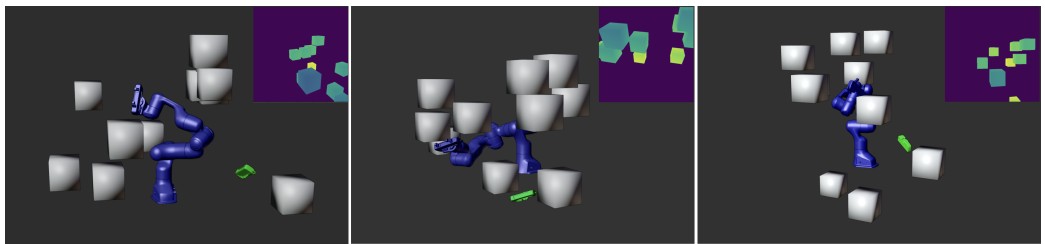

Figure 6: Example scenes and observations for narrow passage problems. The robot is at start state and the goal poses are marked in green.

| Metrics | Success Rate | Jerk ($rad/s^3$) | Motion Time($s$) | T Err($mm$) | Q Err($°$) |
|---|---|---|---|---|---|
| DiffusionSeeder | 52.4% | **65.1** | 1.45 | **0.08** | **0.73** |
| cuRobo | 52.4% | 100.33 | **1.17** | 0.19 | 0.91 |

Table 3: Experiments on 500 narrow passage problems for DiffusionSeeder and cuRobo with partial observations. As DiffusionSeeder is never trained on such scenes, the performance of both methods are similar.

- **Success Rate:** the percentage of collision-free trajectories with the end pose translation error and quaternion error lower than $\delta_t$ and $\delta_r$ respectively.

- **Plan Time:** the time for the planner to generate the trajectories given the observation.

- **Jerk:** the maximum jerk of the flattened trajectories over all time steps and robot joints.

- **Translation Error:** the mean squared error between the achieved end pose position and the target pose position.

- **Quaternion Error:** the cosine distance between the achieved end pose quaternion and desired end pose quaternion.

- **Motion Time:** overall execution time of the trajectories accounting for a robot's mechanical limits, such as maximum velocity, acceleration, and jerk.

## 7.8 Simulation Experiments

We represent scenes with a single depth image. Both cuRobo and DiffusionSeeder use nvblox with the given depth image for collision checking. As we notice the performance of nvblox is sensitive to the image size, we render depth images of size $640 \times 640$ at the test time for nvblox EDSF reconstruction and downsample them to $256 \times 256$ as the input to the diffusion model.

### 7.8.1 Evaluation on Narrow Passage Problems

To evaluate the generalization ability of DiffusionSeeder, we evaluate DiffusionSeeder on narrow passage problems. We generate 500 problems on 50 scenes consisting of 10 cuboid obstacles from the sparrow dataset [35]. Results are shown in Table 3. Both DiffusionSeeder and cuRobo achieve a success rate of 52.4% when given partial observation of depth images. Examples of the problems and the depth observations are shown in Figure 6. As DiffusionSeeder is never trained on such scenes, the performance of DiffusionSeeder is similar to cuRobo. In the future work, we expect to include more diverse scene types in the training of DiffusionSeeder.

| Metrics | Success Rate | Jerk ($rad/s^3$) | Motion Time($s$) | T Err($mm$) | Q Err($°$) |
|---|---|---|---|---|---|
| DiffusionSeeder | 80.6% | 77.0 | **1.26** | **0.57** | **1.73** |
| DiffusionSeeder with Guidance | **81.1%** | **76.7** | 1.27 | 0.60 | 1.77 |

Table 4: Experiments on 1791 test problems for DiffusionSeeder with and without guidance with partial observations. Cost gradient guidance improves the success rate of DiffusionSeeder marginally.

### 7.8.2 Case Study for Scene Occlusion

For robotic motion planning, the same observations may bring different levels of difficulties for different motion planning problems, depending on the information needed for solving the problem. Therefore, to systematically evaluate the occlusion, we conduct a case study on one cubby scene with 50 sampled camera views. We sample 56 problems for this scene. For each observation, we run both DiffusionSeeder and cuRobo on all 56 problems and calculate the average success rate. The success rate of each method for different camera views are summarized in Figure 7. On average of all 50 camera views, DiffusionSeeder achieved a success rate of 65.9% while cuRobo achieved a success rate of 60.2%. From the Figure, the success rate of DiffusionSeeder is comparable with cuRobo when the scene is less occluded (green region), while DiffusionSeeder performs better than cuRobo when the scene is more occluded (yellow region). This indicates DiffusionSeeder has implicit scene completion ability and can plan better when the scene is heavily occluded.

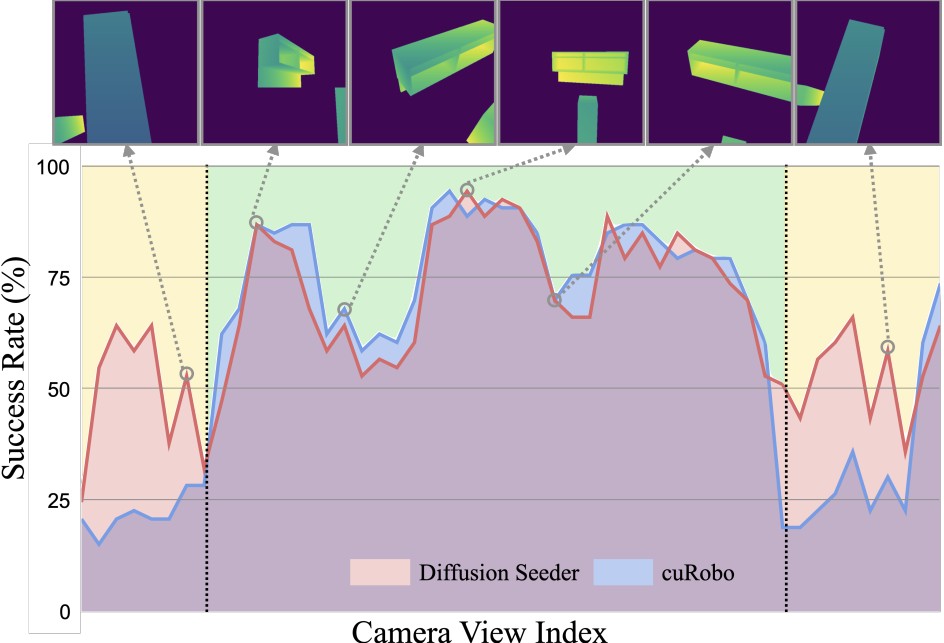

Figure 7: Success rate over 56 problems on a cubby scene for DiffusionSeeder (red) and cuRobo (blue) for 50 different camera views. Depth observation examples are shown on the top. In general, the observations in the yellow region are more occluded while those in the green region are less occluded.

### 7.8.3 Cost Guided Diffusion Sampling

Similar to [27, 28], we evaluate the benefits of guiding diffusion sampling with the cost gradient of the current scene. Specifically, we have

$$\tau_{t-1} \sim \mathcal{N}(\mu_\theta(\tau_t, t) + \alpha \Delta_\tau \mathcal{C}, \Sigma_t) \tag{5}$$

where $t$ is the diffusion denoising time step, $\mu_\theta$ is the reverse diffusion, $\Sigma_k$ is covariance schedule, $\Delta_\tau \mathcal{C}$ is the gradient of the cost with respect to the current sampled trajectory $\mu_\theta(\tau_t, t)$ and $\alpha$ is a weight scalar. As in [27], we implement the cost gradient guidance for $t < k$ and take $n$ gradient steps for each time step $t$. We also notice excluding the guidance for the last denoising step improves the performance.

We evaluate DiffusionSeeder with and without guidance on the 1791 test problems with partial observations. We set $k = 60, n = 20$ and $\alpha = 1$, which achieves the best performance through a parameter sweeping experiment. Results are summarized in Table 4. Notice the results of DiffusionSeeder is different from that reported in Table 1 as a more recent version of cuRobo (v0.7.4) is

used. From the results, the gradient guidance improves the performance marginally. We suspect that the improvement is marginal because DiffusionSeeder is conditioned on the scene and can already output seed trajectories at the low cost region of the scene. We leave how to better incorporate the gradient guidance in the diffusion denoising process to future work.

### 7.8.4 Optimization Steps Analysis

From Table 1, DiffusionSeeder-475 has the highest success rate of 86.2% and DiffusionSeeder-25 has the lowest planning time of 15ms, with a success rate of 85.1%. The trajectory quality increases with $N_{\text{iters}}$, indicated by the decreasing jerk, motion time, pose translation, and quaternion error, at the cost of increasing planning time from $15ms$ to $45ms$. We observe that the success rate doesn't vary significantly with $N_{\text{iters}}$. We hypothesize that this is because the diffusion model is trained on the trajectories optimized for motion planning cost on the ground truth mesh and additionally sees multiple different views of the scene during training. It implicitly minimizes the trajectory optimization cost and can do implicit scene completion from partial observations. Therefore, the seed trajectories are closer to the optimal collision-free trajectory while avoiding collision in unobservable areas and require fewer optimization steps to generate collision-free trajectories. Contrarily, cuRobo optimizes on trajectories generated by a graph-based planner or linear interpolation that are less optimal, requiring more optimization iterations to minimize the motion optimization cost.

### 7.8.5 Trajectory Quality Comparisons

Among all methods, BiTStar has the lowest jerk but also the lowest success rate. MπNet has the highest motion time and target pose tracking error. The maximum jerk of DiffusionSeeder generated trajectories is slightly higher than cuRobo but still far below the robot's mechanical limit. DiffusionSeeder has slightly higher motion time, likely coming from the non-linear trajectories that cuRobo fails to solve. The additional weight $\alpha$ on loss for non-linear trajectories (see Section 4.3) may also contribute to this high motion time of DiffusionSeeder. Both DiffusionSeeder and cuRobo have sub-millimeter target pose translation tracking error and low quaternion tracking error.

### 7.8.6 Full vs Partial Observability

| Metric | BiTStar | | MπNet | | cuRobo v0.6.2 | | | DiffusionSeeder | | | |
|---|---|---|---|---|---|---|---|---|---|---|---|
| Condition | $\tilde{\delta}'$ | $\delta$ | $\delta'$ | $\delta$ | $N_{atp}=1$ | $N_{atp}=10$ | $N_{atp}=100$ | $N_{iters}=25$ | $N_{iters}=50$ | $N_{iters}=100$ | $N_{iters}=200$ |
| Plan Time (s) | 0.81 | 0.45 | 0.39 | 0.54 | 0.046 | 0.065 | 0.068 | **0.014** | 0.016 | 0.020 | 0.027 |
| Success Rate | 61.1% | 18.0% | 62.31% | 48.4% | 87.0% | 99.72% | **99.77%** | 96.6% | 97.0% | 96.6% | 96.7% |
| Jerk ($rad/s^3$) | **55.3** | 59.7 | 101.5 | 106.0 | 110 | 110 | 110 | 113.44 | 97.4 | 80.1 | 71.5 |
| Motion Time (s) | 2.27 | 2.57 | 4.23 | 4.78 | **1.05** | **1.05** | **1.05** | 1.25 | 1.26 | 1.29 | 1.31 |
| Translation Err (mm) | 3.98 | 3.92 | 7.16 | 3.28 | 0.1 | 0.1 | 0.1 | **0.03** | 0.07 | 0.05 | **0.03** |
| Quaternion Err(°) | 13.07 | 0.92 | 3.36 | 2.23 | 0.55 | 0.54 | 0.53 | 1.99 | 1.26 | 0.29 | **0.17** |

Table 5: Evaluation with ground truth mesh for BiTStar, MπNet, cuRobo and DiffusionSeeder. Mean values of each metric over the 1791 test problems are reported. Same as experiments with partial observations, we report results of BiTStar and MπNet with different success threshold and cuRobo with different number of attempts and DiffusionSeeder with different number of optimization iterations.

In this set of experiments, we pass the ground truth mesh for optimizer collision checking for BiTStar, cuRobo, and DiffusionSeeder. For MπNet, we pass point clouds sampled from the ground truth mesh as in MπNet. We calculate each metric for successful trajectories, with the average value summarized in Table 5. From Table 5, BiTStar has a success rate of 61% under success threshold $\tilde{\delta}$, providing a benchmark of the difficulty of the problems. cuRobo-100 has the highest success rate of 99.7%, while DiffusionSeeder-50 has a success rate of 97%. It's worth noting that the diffusion model of DiffusionSeeder still takes partial observations of a depth image while cuRobo takes in full observations, which explains the slightly lower success rate of DiffusionSeeder compared to cuRobo and the consistent success rate across different $N_{\text{iters}}$. The trajectory quality with $N_{\text{iters}}$ shows similar trends to those in partially observed environments. In fully observed environments, cuRobo-100 is still significantly slower than DiffusionSeeder as shown in Figure 3b, where DiffusionSeeder-50 is 6x faster on average, 4x faster on 75th quantile and 22x faster on the 98th quantile. As aforementioned, cuRobo calls a graph based planner when the linear interpolation trajectory fails for

complicated problems, resulting in the significant slow down on obstacle-dense scenes. In contrast, DiffusionSeeder has a consistent planning time over all scene types.

### 7.8.7 Impact of $N_{\text{trajs}}$

We study the performance of DiffusionSeeder with $N_{\text{trajs}}$ in partially observed environments. We set $N_{\text{iters}} = 200$. Results are shown in Table 6. The success rate of DiffusionSeeder increases monotonically upto $N_{\text{trajs}} = 12$, suggesting $N_{\text{trajs}} = 12$ is able to represent the potential modalities sufficiently. The success rate of $N_{\text{trajs}} = 12$ is 1.99x higher than that of $N_{\text{trajs}} = 1$, highlighting the benefits of using the diffusion model to generate multiple diverse seeds, compared to motion planning policies that generate one initial trajectory through multi-step rollout. We notice that the success rate is not increasing monotonically after $N_{\text{trajs}} = 12$. We hypothesize this might be because as $N_{\text{trajs}}$ increases, the ratio of linear trajectories also increases, resulting the optimizer to optimize a more linear trajectory, which is implied by the lower motion time for higher $N_{\text{trajs}}$.

| $N_{\text{trajs}}$ | 1 | 2 | 4 | 8 | 12 | 16 | 32 |
|---|---|---|---|---|---|---|---|
| Plan Time (s) | 0.023 | 0.039 | 0.030 | 0.027 | 0.027 | 0.026 | 0.04 |
| Success Rate | 42.8% | 47.5% | 70.7% | 77.7% | **85.1%** | 82.7% | 84.3% |
| Jerk ($rad/s^3$) | 79.5 | 66.4 | 74.6 | 84.7 | 93.5 | 84.2 | 85.5 |
| Motion Time (s) | 1.32 | 1.37 | 1.33 | 1.30 | 1.30 | 1.24 | 1.24 |
| Translation Err (mm) | 0.35 | 0.27 | 0.36 | 0.51 | 0.50 | 0.41 | 0.45 |
| Quaternion Err (°) | 0.68 | 0.59 | 0.78 | 0.90 | 1.03 | 0.86 | 1.71 |

Table 6: Evaluation with partial observations of a 640×640 depth image for DiffusionSeeder with different $N_{trajs}$. Mean values of each metric over the 1791 test problems are reported. The performance increases monotonically until $N_{trajs} = 12$.

### 7.8.8 Impact of Number of Views

DiffusionSeeder can take in multiple depth images by design. Though we only train the model on one depth image, we can provide multiple depth images at the inference time. We set $N_{\text{iters}} = 200$ and $N_{\text{trajs}} = 12$ for DiffusionSeeder and provide 1, 2, and 5 depth images. As shown in Table 7, increasing number of camera views does not improve the performance, likely because DiffusionSeeder is only trained on one depth image.

| Num of Views | 1 | 2 | 5 |
|---|---|---|---|
| Plan Time (s) | 0.027 | 0.025 | 0.024 |
| Success Rate | 85.1% | 84.5% | 83.8% |
| Jerk ($rad/s^3$) | 93.5 | 93.1 | 88.2 |
| Motion Time (s) | 1.30 | 1.23 | 1.23 |
| Translation Err (mm) | 0.50 | 0.81 | 0.69 |
| Quaternion Err (°) | 1.03 | 1.75 | 1.62 |

Table 7: Evaluation of DiffusionSeeder with different number of 640×640 depth images. Mean values of each metric over the 1791 test problems are reported.

### 7.8.9 Performance of DiffusionSeeder without cuRobo

We evaluate the performance of DiffusionSeeder without integration of cuRobo (i.e., no trajectory optimization). We sample $N_{\text{trajs}}$ from the diffusion model and choose the trajectory that has the lowest translation error as the final solution. We do collision checking with the ground truth mesh and check if the translation error and quaternion error is below the success threshold to calculate the success rate.

Results are summarized in Table 8. As $N_{\text{trajs}}$ increases, the success rate increases monotonically. The collision-free rate, which is consistent with respect to the $N_{\text{trajs}}$, is the upper limit of the success rate of the diffusion model as collision-free trajectories can have large end pose errors. The success rate

| $N_{\text{trajs}}$ | S.R. | CFR | Jerk ($rad/s^3$) | M.T. (s) | T Err (mm) | Q Err (°) |
|---|---|---|---|---|---|---|
| 8 | 13.2% | 38.1% | 291.5 | 1.36 | 3.37 | 2.55 |
| 12 | 15.4% | 37.9% | 289.9 | 1.36 | 3.28 | 2.60 |
| 16 | 17.2% | 37.8% | 286.2 | 1.37 | 3.22 | 2.64 |
| 32 | 21.1% | 38.0% | 280.6 | 1.40 | 3.13 | 2.62 |
| 64 | 22.4% | 37.7% | 282.5 | 1.39 | 2.86 | 2.72 |
| 128 | 26.2% | 38.0% | 273.0 | 1.42 | 2.70 | 2.66 |
| 256 | 27.6% | 38.0% | 269.1 | 1.43 | 2.47 | 2.76 |
| 512 | 29.3% | 38.0% | 268.1 | 1.43 | 2.29 | 2.79 |
| 1024 | 31.5% | 37.9% | 268.4 | 1.43 | 2.13 | 2.88 |

Table 8: Performance of DiffusionSeeder without cuRobo. We report the Success Rate (S.R.), the collision-free Rate (CFR), the maximum Jerk, the Motion Time (M.T.), the Translation Error (T Err) and the Quaternion Error (Q Err). The collision-free rate is defined as the number of the collision-free trajectories among all $N_{\text{trajs}}$ sampled trajectories.

of the diffusion model is much higher than M$\pi$Net (Table 1), indicating that the diffusion model is able to generate environment dependent seeds. However, the success rate is much lower than that of DiffusionSeeder integrated with cuRobo, highlighting the benefits of optimizing the diffusion seeds with explicit collision checking using nvblox and other cost terms. In addition, diffusion generated trajectories tend to have a high maximum jerk of around 270 $rad/s^3$, which is significantly reduced by cuRobo optimization to around 106 $rad/s^3$.

## 7.9 Real Robot Evaluation

We use a Realsense D435 depth camera mounted opposite to the robot to obtain depth images. The robot is segmented and removed from the depth images and then sent to DiffusionSeeder and nvblox. None of the environment setups are part of our training dataset, in addition the chosen camera view causes severe occlusions in the environment when obstacles are present as seen in Figure 1. Hence we used $N_{\text{iters}} = 100$ for DiffusionSeeder as using 50 did not get us high-quality solutions. For cuRobo, we use the default number of iterations (475) provided in their code base and have $N_{\text{atp}} = 10$. For both methods, we run the Franka Panda with a joint impedance controller so that collisions in the world do not damage the robot. In addition, we set the maximum allowed velocity to 65% of the robot's limits across the methods to avoid any catastrophic damage due to occlusions in the world.

