# OpenReview forum: "DiffusionSeeder: Seeding Motion Optimization with Diffusion for Rapid Motion Planning"
_robot-learning.org/CoRL/2024/Conference — CoRL 2024_

### Official Review · Reviewer_oM7J · 2024-07-16
**Essentially a big data approach.**

**Originality:** 3
**Technical Quality:** 3
**Clarity Of Presentation:** 3
**Potential Impact:** 3
**Recommendation:** 3
**Confidence:** 4

**Review:**

**Novelty & Significance**

1. The method is essentially an engineering work that vision-conditional diffusion models initialize downstream trajectory optimizers, e.g., curobo optimizers. This is essentially without the prior and likelihood blending in [1]. The contribution is, therefore, judged by the significance of the experiment results.

2. The loss Eq. 2 needs proper motivations since matching end-effector poses only basically discard self-collision and collision information from the planner data. Moreover, FK is inherently highly non-linear; matching poses noise via FK amplifies noises. Perhaps it works due to having an enormous dataset, but I cannot see why this helps training diffusion.

3. Although diversity is not a focus of this work, I believe that with such a big dataset, the diffusion prior can generate very diverse and high-quality modes for downstream trajectory optimization.

**Quality & Clarity**

1. The method descriptions and figures are well-written and easy to read.

2. The experiment outline has good and diverse test environments. The baselines are also adequate. However, since this work is very close to [1] (although [1] does not condition on image/depth observations), this work should still compare to [1] by reusing the trained diffusion prior and curobo [2] planner with the blending in Algorithm 1 of [1]. This can be considered an ablation study of with vs. without blending diffusion prior update and likelihood (i.e., planner) gradient update.

-------------------------
**Post Rebuttal Comments**

I would like to thank the authors for their efforts in addressing my concerns. I raise my score to Weak Accept.


**References**

[1] Carvalho, Joao, et al. "Motion planning diffusion: Learning and planning of robot motions with diffusion models." 2023 IEEE/RSJ International Conference on Intelligent Robots and Systems (IROS). IEEE, 2023.

[2] Sundaralingam, Balakumar, et al. "Curobo: Parallelized collision-free robot motion generation." 2023 IEEE International Conference on Robotics and Automation (ICRA). IEEE, 2023.

**Quality Of The Limitations Section:**

3

**Questions For Rebuttal:**

1. Why does DiffusionSeed have high jerks compared to a sampling-based approach such as BitStar?
2. Please compare DiffusionSeeder to [1], with the focus on comparing two paradigms of coupling vs. decoupling diffusion prior update and likelihood (i.e., planner) gradient update, to see if the best possible method configuration of diffusion prior for motion planning.

**Robotics Focus:**

4

**Summary Of Paper:**

DiffusionSeeder uses a conditional Denoising Diffusion Probabilistic Model (DDPM) to generate diverse, high-quality seed trajectories from initial depth images of the environment. These seed trajectories are then optimized using cuRobo, a GPU-accelerated motion optimizer. The key claim is that DiffusionSeeder accelerates motion planning and increases success rates in complex environments with image observations by providing high-quality seed trajectories for trajectory optimization, trained on big datasets. The evaluation involves both simulated and real-world experiments using the Franka robot. In simulations, DiffusionSeeder is tested on 3M scenes from the MπNet dataset, comparing performance with BiTStar, MπNet, and cuRobo.

**Summary Of Recommendation:**

This works has clear motivations and clear engineering description. However, I am not totally convinced on the experiment statistics. I will increase my score when additional experiment results added after the rebuttal.

---

### Official Review · Reviewer_QoEm · 2024-07-21
**Good work with broad experiments**

**Originality:** 3
**Technical Quality:** 3
**Clarity Of Presentation:** 4
**Potential Impact:** 3
**Recommendation:** 3
**Confidence:** 4

**Review:**

This work proposes to train a diffusion model for a large-scale motion planning problem. The output paths of the model become the initial trajectories for a trajectory optimizer.

The method supports the raw depth image input and utilizes the strength of ViT. The diffusion part uses the DDPM model.

The paper's method is not very novel while the experiments are solid. The paper demonstrates it is possible to make an end-to-end motion planning from raw observation.

However, despite the variance of the input observation, the generated paths have some similar patterns, for example, the manipulator's end effector goes up and down. I think it reflects the planning problems' similarity in the dataset. While real-world problems may include more cases, e.g. the narrow passage problem in Riemannian Motion Policies [2].

Reference missing. There exists a concurrent diffusion work [1] for motion planning on MpiNet's dataset.

[1] EDMP: Ensemble-of-costs-guided Diffusion for Motion Planning

[2] Riemannian Motion Policies

**Quality Of The Limitations Section:**

3

**Questions For Rebuttal:**

1. I noticed the experiments in the video are either the end effector going up and going down, or the end effector going into the cabinet. Could this method work for more challenging narrow passage problems? For example, multiple links go into the cabinet.

**Robotics Focus:**

4

**Summary Of Paper:**

This paper propose to use diffusion method to generate conditional initial trajectories for trajectory optimization.

**Summary Of Recommendation:**

This work propose to use ViT conditioned difffusion to generate initial planning paths and demonstrates its effectiveness on a large dataset. The method can have a good impact for further large scaled robot learning.

---

### Official Review · Reviewer_kGxy · 2024-07-21
**DiffusionSeeder: Seeding Motion Optimization with Diffusion for Rapid Motion Planning**

**Originality:** 4
**Technical Quality:** 4
**Clarity Of Presentation:** 4
**Potential Impact:** 3
**Recommendation:** 4
**Confidence:** 4

**Review:**

Strengths:
- The overall presentation is clear, easy to read, and understand.
- The motivation of the work is convincing as it is closely connected with cuRobo.
- The proposed idea is novel, using the output of diffusion models as good-quality seed trajectories to increase the likelihood of finding a solution trajectory.
- The literature review appears comprehensive, as it covers both classical methods and recent learning-based methods.
- The experimental results are promising, and the evaluations are exhaustive. They include various baselines, and diverse aspects have been validated.

Weaknesses:
- Occlusion evaluation is not clearly demonstrated, making the approach to dealing with partial observability questionable. Since this is one of the major contributions, this aspect must be systematically evaluated. How severe occlusion can the method handle?
- A new loss function (Equation (2)) is proposed, but its effectiveness is not evaluated. Therefore, it is unclear why the new loss function is more beneficial than the original one.

**Quality Of The Limitations Section:**

3

**Questions For Rebuttal:**

- Please address the two comments in the weaknesses above.
- In Figure 1’s caption, “nvblox [1] online from the depth images …” What does "online" mean here? I thought the overall pipeline computes a trajectory offline, so this part was confusing.
- Figure 2 does not include the “time step” latent vector.
- In Section 5.1: 575 dim -> 576 dim
- At the end of Section 5.1: “831-dimensional conditional vector for each layer.” Which layer does this refer to?
- In Figure 4, mark the goals. Otherwise, it’s hard to judge whether the problem is easy or hard.

**Robotics Focus:**

4

**Summary Of Paper:**

The paper proposes diffusion model-based trajectory generation, which can be used to improve planning efficiency and the success rate of trajectory optimization in complex environments. The method additionally addresses handling partial observability by considering environment conditioning with a depth image obtained by a fixed camera. The proposed method shows promising results compared to various baselines in both planning time and success rate.

**Summary Of Recommendation:**

The paper's contribution appears significant, and the presentation and technical details are clear and sufficient.

---

### Author Rebuttal · Authors · 2024-08-09

We thank all reviewers and the AC for their valuable feedback! Based on the feedback, we performed the following additional experiments and changes. The revised draft is uploaded.

* We have added experiments on more challenging narrow passages problems in **Appendix 8.8.1**. Despite DiffusionSeeder has never been trained on such scenes, it achieves the same success rate as cuRobo under partial observations.

* We have added experiments on incorporating the cost gradient guidance from the prior work [1] to DiffusionSeeder in **Appendix 8.8.3**, where we see a marginal performance improvement of 0.5% success rate by adding this gradient guidance compared to without the gradient guidance. We suspect that this is because DiffusionSeeder is conditioned on the scene and can already output seed trajectories at the low cost region of the scene. Therefore DiffusionSeeder doesn’t benefit much from the gradient guidance.

* We have added explanations on the higher jerk of DiffusionSeeder compared to BiTStar in **Section 6.1.1**, where we show the higher jerk comes from the faster trajectories generated by the DiffusionSeeder and that re-timed BiTStar trajectories have a 21% lower jerk but are 27% slower than DiffusionSeeder generated trajectories.

* We have added clarifications on the loss design in **Section 5.2**: “*we use FK to obtain the position of many points sampled on the robot's geometry across all links. We calculate the distance between the predicted positions of these points and the labels (using FK on joint state) as the loss. This loss gives a more direct representation of the error in the Cartesian Space for the whole robot.*”

* In addition, we have added a case study on the scene occlusions in **Appendix 8.8.2**, where we show DiffusionSeeder has a better performance on heavily occluded scenes than cuRobo and achieves a higher success rate on average.

[1]  Carvalho, Joao, et al. "Motion planning diffusion: Learning and planning of robot motions with diffusion models." 2023 IEEE/RSJ International Conference on Intelligent Robots and Systems (IROS). IEEE, 2023.

---

### Decision · Program_Chairs · 2024-09-04

**Decision:**

Accept

**Comment:**

The paper proposes the use of the diffusion method to generate conditional initial trajectories for trajectory optimization. Overall, the paper is well-written and easy to follow. However, it is necessary to demonstrate the proposed work in more challenging narrow passages to validate its capability. Furthermore, a comparison against relevant baselines such as DiffusionSeede is necessary to validate the contribution of the proposed approach over prior work. Additionally, the results should be clearly explained, including why the proposed method has more jerks in the trajectories, and the design of the objective function also needs to be motivated.

Update: The revised version has addressed the reviewer's comments to a good extent.